# Assessment of the Effectiveness of Zone 1-Landing Hybrid TEVAR by Comparing Its Outcomes with Those of Zone 2-Landing Hybrid TEVAR

**DOI:** 10.3390/jcm12165326

**Published:** 2023-08-16

**Authors:** Tomoaki Kudo, Toru Kuratani, Yoshiki Sawa, Shigeru Miyagawa

**Affiliations:** 1Department of Cardiovascular Surgery, Graduate School of Medicine, Osaka University, Suita 565-0871, Osaka, Japan; 2Department of Minimally Invasive Cardiovascular Medicine, Graduate School of Medicine, Osaka University, Suita 565-0871, Osaka, Japan

**Keywords:** thoracic endovascular aortic repair, hybrid arch repair, aortic arch aneurysm, dissecting, aortic aneurysm, extra-anatomical bypass, intracranial blood flow

## Abstract

**Background:** Hybrid thoracic endovascular aortic repair (TEVAR) without median sternotomy is increasingly being performed in high-risk patients with aortic arch disease. The outcomes of hybrid TEVAR were reported to be worse with a more proximal landing zone. This study aims to clarify the effectiveness of zone 1-landing hybrid TEVAR by comparing the outcomes of zone 2-landing hybrid TEVAR. **Methods:** From April 2008 to October 2020, 213 patients (zone 1: zone 1-landing hybrid TEVAR, *n* = 82, 38.5%; zone 2: zone 2-landing hybrid TEVAR, *n* = 131, 61.5%) were enrolled (median age, 72 years; interquartile range [IQR], 65–78 years), with a median follow-up period of 6.0 years (IQR, 2.8–9.7 years). **Results:** The mean logistic EuroSCORE was 20.9 ± 14.8%: the logistic EuroSCORE of the zone 1 group (23.3 ± 16.1) was significantly higher than that of the zone 2 group (19.3 ± 12.4%, *p* = 0.045). The operative time and hospital stay of the zone 1 group were significantly longer than those of the zone 2 group. On the other hand, the in-hospital and late outcomes did not differ significantly between the two groups. There were no significant differences in cumulative survival (66.8% vs. 78.0% at 10 years, Log-rank *p* = 0.074), aorta-related death-free rates (97.6% vs. 99.2% at 10 years, Log-rank *p* = 0.312), and aortic event-free rates (81.4% vs. 87.9% at 10 years, Log-rank *p* = 0.257). **Conclusions:** Zone 1- and 2-landing hybrid TEVAR outcomes were satisfactory. Despite the high procedural difficulty and surgical risk, the outcomes of zone 1-landing hybrid TEVAR were equal to those of zone 2-landing hybrid TEVAR. If the surgical risk is high, zone 1-landing hybrid TEVAR should not be avoided.

## 1. Introduction

The outcomes of hybrid thoracic endovascular aortic repair (TEVAR) have been reported to be worse when performed in more proximal landing zones [1,2]. There are many reports of zone 0- and zone 2-landing hybrid TEVAR for aortic arch diseases, but few reports of zone 1-landing hybrid TEVAR [2,3,4,5,6,7]. This is because an extra-anatomical bypass (right axillary artery [AxA]–left common carotid artery [LCCA]–left AxA bypass) and intracranial blood flow from the brachiocephalic artery (BCA) are required for zone 1-landing hybrid TEVAR. In our previous study, zone 1-landing hybrid TEVAR with extra-anatomical bypass did not influence the total intracranial flow [8]. This study aims to clarify the effectiveness of zone 1-landing hybrid TEVAR by comparing its outcomes with those of zone 2-landing hybrid TEVAR.

## 2. Materials and Methods

### 2.1. Ethics Statement and Study Design

This single-center, retrospective, observational cohort study was conducted in accordance with the Declaration of Helsinki, and the protocol was approved by the Ethics Committee of Osaka University School of Medicine. Informed consent was obtained from all patients before the procedures.

### 2.2. Study Population

Of the 752 patients who had undergone TEVAR for aortic arch pathologies between April 2008 and October 2020, 213 who had undergone zone 1- (Figure 1A) and zone 2- (Figure 1B) landing hybrid TEVARs were enrolled. The mean follow-up period was 6.0 years (interquartile range [IQR], 2.8–9.7 years). None of the patients were lost to follow-up, and all patient data were available.

### 2.3. Indications

The indications for surgical intervention for zone 1 and zone 2 hybrid TEVARs were as follows: aneurysm; aortic diameter expansion by ≥5 mm in 6 months; a maximum aortic diameter > 55 mm; aortic rupture; saccular aneurysm of any size; dissection; malperfusion syndrome; and an initial aortic diameter > 40 mm in TBAD, in addition to the indications for an aneurysm.

### 2.4. Treatment Strategy

The following anatomy of aorta conditions were satisfied before performing zone 1- and zone 2-landing TEVARs: proximal LZ; diameter: ≤42 mm and length > 15 mm, and proximal stent graft; size ≤ 46 mm and oversizing rate: 10–20%. We selected a stent graft with the same oversizing for aneurysm and dissection.

Debranching procedures were as follows: zone 1-landing hybrid TEVAR; the patients underwent an extra-anatomical bypass from the right AxA to the LCCA and left AxA using a T-shaped, ringed, 8-mm expanded polytetrafluoroethylene graft, and zone 2-landing hybrid TEVAR; the patients underwent intentional covering of the left subclavian artery (LSA) without revascularization or extra-anatomical bypass (right AxA to left AxA and LCCA to left AxA). TEVAR was performed after the bypass within the same surgery. Since 2011, we have blocked the native forward flow of the blood vessel through which the device passes before TEVAR to prevent stroke due to thromboembolism. From 2012 onward, to prevent type 2 endoleak, we embolized the blood vessels using coils or ligated directly. Cerebrospinal fluid drainage was performed on patients with stent graft distal edge Th ≤ 10. The patients with atheroma grade ≥ 3 at the proximal LZ underwent TEVAR using a filter device from 2015, as previously described.

### 2.5. Follow-Up

Follow-ups were performed during regular patient visits. The patients were followed up with at least once every 3 months for the first year and every 6 months or 1 year thereafter at our hospital for MDCT examination. Patient death was confirmed via telephone interviews with the patients’ families.

### 2.6. Outcome Criteria

The primary outcomes of interest were (1) aortic events; known or suspected events, such as stroke, aneurysm enlargement ≥ 5 mm in diameter, or any cases of endoleak; stent graft migration; aortic rupture; aortic dissection; and prosthetic infection; and (2) aorta-related death (defined as death due to adverse events secondary to aortic pathologies). False lumen (FL) reperfusion was defined as persistent FL flow from a proximal entry tear at dissecting aortic aneurysm. We defined a stroke as having neurologic symptoms.

### 2.7. Statistical Analyses

Results are expressed as mean ± standard deviation and median (IQR) according to the normality of the distribution, assessed using the Shapiro–Wilk test, and compared using the Mann–Whitney U test. Categorical variables are presented as counts and percentages using the chi-square test or Fisher’s exact test. The curves for overall survival and freedom from aorta-related death and aortic events were estimated using the Kaplan–Meier product-limiting method and compared using the log-rank test. Estimates are provided with 95% CIs. All *p* values were 2-sided, and statistical significance was set at *p* < 0.05. All statistical analyses were performed using JMP statistical software, version 16.0.0 for MacOS X (SAS Institute Inc., Cary, NC, USA).

## 3. Results

### 3.1. Preoperative Patient Characteristics and Measurements

Preoperative patient characteristics and measurements are listed in Table 1. The median patient age at surgery was 72 years (IQR, 65–78 years); 43 (20.2%) patients were older than 80 years, and 42 (19.7%) patients were females. The patient age was significantly higher in the zone 1 group (73 years, IQR: 68–79 years) than in the zone 2 group (71 years, IQR: 61–77 years; *p* = 0.022). The pathologies were attributed to aneurysms in 144 (67.6%) patients and dissection in 69 (32.4%). The number of patients who had type B aortic dissection was significantly higher in the zone 2 group (*n* = 58, 44.3%) than that in the zone 1 group (*n* = 11, 13.4%; *p <* 0.001). Eleven (5.2%) patients had a history of median sternotomy before this study. Thirty-three (15.5%) patients underwent emergency procedures. The mean logistic EuroSCORE was 20.9 ± 14.8%; the logistic EuroSCORE of the zone 1 group (23.3 ± 16.1%) was significantly higher than that of the zone 2 group (19.3 ± 12.4%, *p* = 0.045).

The median maximum diameter of the aorta was 51 mm (IQR, 44–60 mm). The mean diameter of proximal LZ was 31.3 ± 3.5 mm; 32.4 ± 3.1 mm in the zone 1 group and 30.7 ± 3.6 mm in the zone 2 group (*p* = 0.001), respectively. The median length of proximal LZ was 20.0 mm (IQR, 16.9–24.0 mm); 20.0 mm (IQR, 16.5–25.0 mm) in the zone 1 group and 19.0 mm (17.0–24.0 mm) in the zone 2 group (*p* = 0.462), respectively. The mean diameter of the distal LZ was 27.1 ± 4.2 mm.

### 3.2. Procedure and Stent Grafts

The procedures and stent grafts used are listed in Table 2. All procedures were successful, and the median operating time was 160 min (IQR, 130–204 min): 191 min (IQR, 161–220 min) in the zone 1 group and 140 min (IQR, 112–171 min) in the zone 2 group; the operating time in the zone 1 group was longer than that in the zone 2 group (*p <* 0.001).

Proximal devices such as TAG and CTAG (Gore Medical, Flagstaff, AZ, USA) were used in 162 (76.1%) patients, Zenith TX2 and TXD (Cook Medical, Bloomington, IN, USA) in 17 (8.0%), Relay Plus and Relay NBS (Terumo Aortic, Inchinnan, Scotland, UK) in 26 (12.2%), and Talent and Valiant (Medtronic Inc., Minneapolis, MN, USA) in 8 (3.8%). The median size of the proximal stent graft was 36 mm (IQR, 34–38 mm); 37 mm (IQR, 34–40 mm) in the zone 1 group and 34 mm (IQR, 34–37 mm) in the zone 2 group (*p* = 0.004). The median oversizing rate of the proximal devices was 14% (IQR, 10–18%); 13% (IQR, 10–17%) in the zone 1 group and 14% (IQR, 10–19%) in the zone 2 group (*p* = 0.154). The median thoracic spine (Th) at the distal edge of the stent graft was 8 (IQR 7–9).

### 3.3. In-Hospital Outcomes

The in-hospital outcomes are presented in Table 3. The median postoperative hospital stay was 10 days (IQR, 8–16 days). The 30-day mortality rate was 0.5% (*n* = 1). The hospital mortality rate was 0.9% (*n* = 2); one (0.5%) patient in the zone 1 group had an abdominal embolic event and one (0.5%) had an aneurysm rupture due to a type 1a endoleak. Two (1.0%; one [1.2%] in the zone 1 group and one [0.8%] in the zone 2 group) patients experienced a stroke, and two (1.0%; one [1.2%] in the zone 1 group and one [0.8%] in the zone 2 group) had spinal cord injury. Endoleaks were reported in 11 (5.2%) patients, including type 1a (*n* = 3; 1.4%), type 1b (*n* = 1; 0.5%), and type 1c endoleaks (*n* = 7; 3.3%). Aortic events and endoleaks did not differ significantly between the groups. Of the two patients with type 1a endoleaks, one was treated successfully with conventional arch repair and the other experienced an aneurysm rupture. One patient with a type 1b endoleak underwent TEVAR. All type 1c endoleaks were detected based on LSA. We embolized the LSA using the coils, and the type 1c endoleak disappeared.

### 3.4. Late Outcomes

The late outcomes are summarized in Table 4. Late deaths were reported in 38 (17.8%) patients, including deaths due to stent graft infection (*n* = 1), cardiac events (*n* = 3), cerebrovascular diseases (*n* = 6), malignant diseases (*n* = 10), and infectious diseases (*n* = 17). Twelve (5.6%) patients had late aortic events: one (0.5%) had RTAD, three (1.4%) had distal stent graft-induced new entry (SINE), one (0.5%) had a prosthetic infection, four (1.9%) had type 1a endoleak, and two (0.9%) had type 1b endoleaks (*n* = 1, 0.5%).

Figure 2A shows the 10-year rate of cumulative survival was 73.6% (95% CI: 64.7–80.9%). The survival rates at 10 years in the zone 1 and zone 2 groups were 66.8% (95% CI: 51.9–79.0%) and 78.0% (95% CI: 66.8–86.2%), respectively, with no significant differences (log-rank *p* = 0.074) (Figure 3A).

Figure 2B shows the aorta-related death-free rate at 10 years was 98.6% (95% CI: 95.7–99.5%). The 10-year event-free rates for the zone 1 and zone 2 groups were 97.6% (95% CI: 90.7–99.4%) and 99.2% (95% CI: 94.8–99.9%), respectively, with no significant differences (log-rank *p* = 0.312) (Figure 3B).

Figure 2C shows the aortic event-free rates at 1, 3, 5, and 10 years were 92.0% (95% CI: 87.5–95.0%), 89.4% (95% CI: 84.5–93.0%), 86.7% (95% CI: 81.1–90.9%), and 85.5% (95% CI: 79.4–90.1%), respectively. The 10-year event-free rates for the zone 1 and zone 2 groups were 81.4% (95% CI: 69.2–89.4%) and 87.9% (95% CI: 80.4–92.8%), respectively, with no significant difference (log-rank *p* = *0.257*) (Figure 3C).

### 3.5. Comparison of the Outcomes in Aortic Pathologies

Comparisons of the outcomes for aneurysms and dissections between the two groups are presented in Appendix A. For both aneurysms and dissections, the preoperative patient characteristics did not differ significantly between the two groups. The operative time and hospital stay in the zone 1 group were significantly longer than those in the zone 2 group. In contrast, in-hospital and late outcomes did not differ significantly between the two groups. In addition, the aorta-related death and aortic-event-free rates in aneurysms and dissections did not differ significantly between the two groups (Appendix A).

## 4. Discussion

It is a well-known fact that total arch replacement (TAR) is the first-choice treatment method for aortic arch diseases [9,10,11,12,13]. However, despite recent developments in surgical techniques and the perioperative management of aortic arch diseases, TAR, which requires sternotomy, hypothermic circulatory arrest, selective cervical arterial perfusion, and cardiopulmonary bypass, remains a high-risk procedure, particularly in older adults with significant comorbidities [14,15,16,17]. Some previous studies have reported that high-risk patients aged > 75 years have significantly lower in-hospital mortality, and hybrid TEVAR improves short- and mid-term outcomes, particularly in high-risk patients [18,19]. Some recent studies reported that the long-term results of hybrid TEVAR are equivalent to those of TAR [20,21,22,23]. However, the outcome of TAR is better than that of hybrid TEVAR in terms of aortic event-free and aortic reintervention-free rates [21,23]. Hybrid TEVAR has been established as the minimally invasive first-line treatment for aortic arch diseases at our institution since 2008 [24,25,26].

Zone 0-landing hybrid TEVAR requires a median sternotomy; therefore, zone 1- and zone 2-landing hybrid TEVARs should be recommended from the viewpoint of surgical invasiveness. In our previous study, zone 0-landing hybrid TEVAR performed relatively well. The stroke (1.5%), 30-day mortality (0.5%), 10-year aortic event-free (94.9%), and 10-year aorta-related death-free (82.3%) rates of zone 0-landing hybrid TEVAR were equal to those of zone 1- and zone 2-landing hybrid TEVARs. However, some studies have reported that the outcomes of hybrid TEVAR were worse in the more proximal landing zone. Zone 0 had an increased 30-day mortality rate compared to zone 1 (9.3% in zone 0 vs. 3.7% in zone 1), and zone 1 had an increased rate of ischemic stroke compared to zone 2 (7.7% in zone 1 vs. 6.6% in zone 2) [1,2]. However, as previously reported, zone 0 landing hybrid TEVAR is undoubtedly advantageous for avoiding the bird-beak configuration [25,27]. There is also a zone 0-landing hybrid TEVAR with the chimney technique, but as previously reported, a type 1a endoleak may occur; therefore, it is not recommended except in severe cases [28,29,30].

In the present study, both groups achieved satisfactory in-hospital and late outcomes. In-hospital mortality was 0.5% (1.2% in zone 1 vs. 0% in zone 2), and over 95% of the patients were discharged. The stroke rate was not significantly different between the groups (0.9%, 1.2% in zone 1 vs. 0.8% in zone 2, *p* = 1.00) and is superior to that reported in other studies. Regarding cerebral blood flow, our previous study showed that zone 1 and zone 2-landing hybrid TEVARs do not affect the total intracranial blood flow. In addition, this study revealed that the 10-year aortic event-free and aorta-related death-free rates were 98.6% (97.6% in zone 1 vs. 99.2% in zone 2, Log-rank *p* = 0.312) and 85.5% (81.4% in zone 1 vs. 87.9% in zone 2, Log-rank *p* = 0.257), respectively, which were superior to those reported in other papers on hybrid TEVAR.

It is an undeniable fact that anatomical consideration is important in stent graft therapy. In this study, zone 1-landing hybrid TEVAR had similar outcomes to zone 2-landing hybrid TEVAR, despite the high procedure difficulty and surgical risk. Therefore, zone 1-landing hybrid TEVAR has more outstanding merit in older adults, who cannot be treated with zone 2-landing hybrid TEVAR, than zone 0 landing hybrid TEVAR.

### Limitation

This study has some limitations. First, this was a retrospective study, and some patients had relatively short follow-up periods. Second, the sample size was small for both groups, and more cases are required for further analysis. Therefore, a prospective multicenter study with long-term follow-up is required to confirm our findings.

## 5. Conclusions

The outcomes of zone 1- and zone 2-landing hybrid TEVAR were satisfactory. Zone 0-landing hybrid TEVAR is higher invasive. The high-risk patient with an unsuitable landing zone at zone 2 should not be discouraged from zone 1-landing hybrid TEVAR.

## Figures and Tables

**Figure 1 jcm-12-05326-f001:**
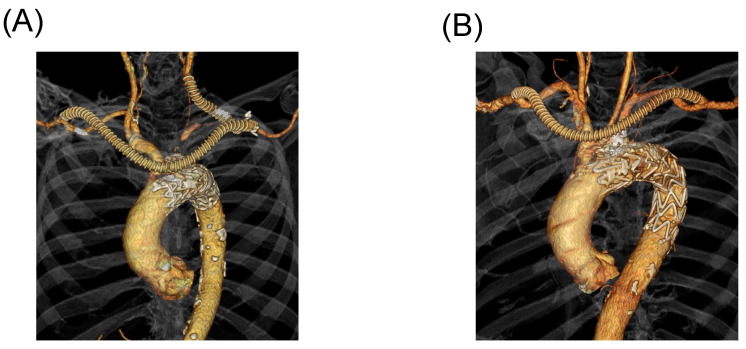
(**A**) Zone 1-landing hybrid TEVAR (zone 1). (**B**) Zone 2-landing hybrid TEVAR (zone 2).

**Figure 2 jcm-12-05326-f002:**
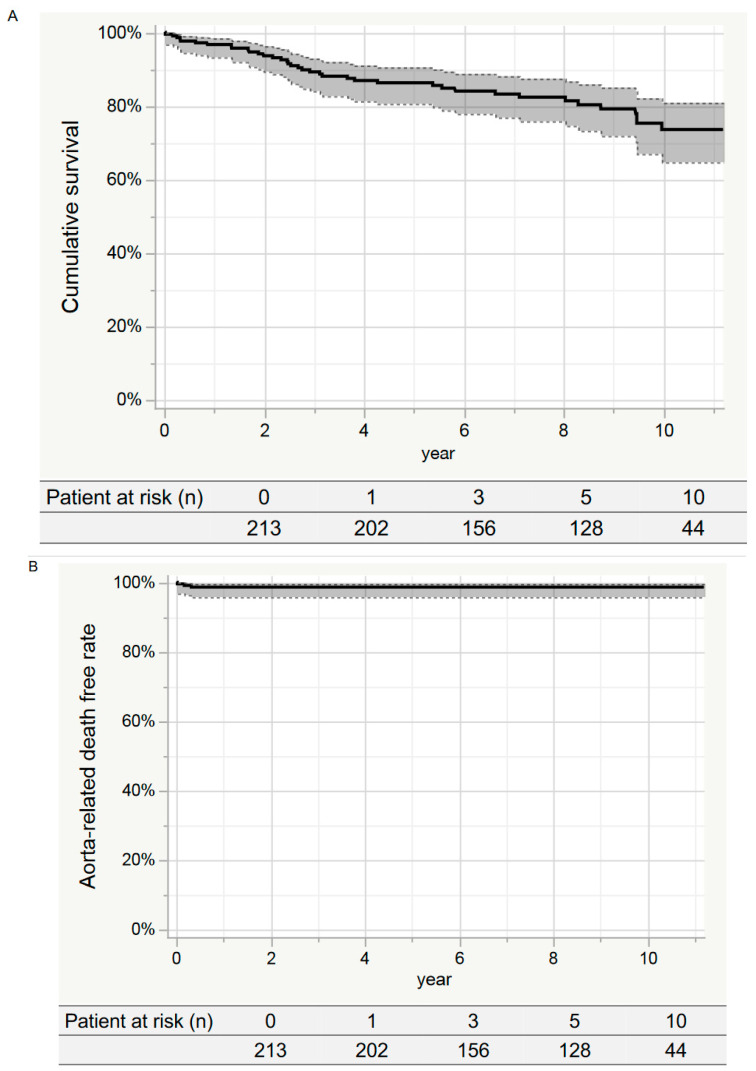
Kaplan–Meier curve in the entire study group. (**A**) Cumulative survival. The cumulative survival rate at 10 years was 73.6% (95% CI: 64.7–80.9%). (**B**) Aorta-related death-free rates. The event-free rate at 10 years was 90.8% (95% CI: 77.2–96.7%). (**C**) Aortic event-free rate. The event-free rate at 10 years was 85.5% (95% CI: 79.4–90.1%).

**Figure 3 jcm-12-05326-f003:**
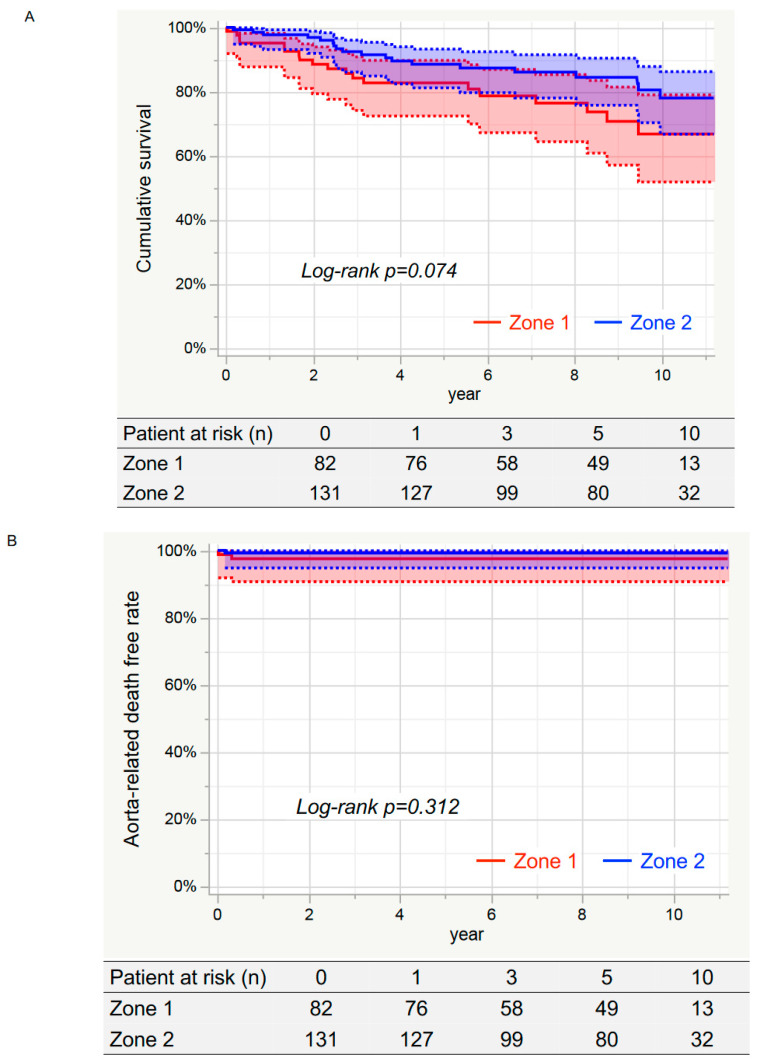
Kaplan–Meier curve in the each groups. (**A**) Cumulative survival. The 10-year survival rates in the zone 1 and zone 2 groups were 66.8% (95% CI: 51.9–79.0%) and 78.0% (95% CI: 66.8–86.2%) (*Log-rank p* = 0.074), respectively. (**B**) Aorta-related death-free rates. The event-free rates at 10 years for the zone 1 and zone 2 groups were 97.6% (95% CI: 90.7–99.4%) and 99.2% (95% CI: 94.8–99.9%) (*Log-rank p* = 0.312), respectively. (**C**) Aortic event-free rates. The event-free rates at 10 years for the zone 1 and zone 2 groups were 81.4% (95% CI: 69.2–89.4%) and 87.9% (95% CI: 80.4–92.8%) (*Log-rank p* = 0.257), respectively.

**Table 1 jcm-12-05326-t001:** Preoperative patient characteristics and measurements.

	All *n* = 213	Zone 1 *n* = 82 (38.5%)	Zone 2 *n* = 131 (61.5%)	*p* Value
Age (years)	72 (65–78)	73 (68–79)	71 (61–77)	0.022
Age ≥ 80 years, *n* (%)	43 (20.2)	19 (23.2)	24 (18.3)	0.391
Female, *n* (%)	42 (19.7)	13 (15.9)	29 (22.1)	0.262
Aortic pathologies				
TAA, *n* (%)	144 (67.6)	71 (86.6)	73 (55.7)	<0.001
DAA, *n* (%)	69 (32.4)	11 (13.4)	58 (44.3)	
Acute and subacute, *n* (%)	21 (9.9)	4 (4.9)	17 (13.0)	0.725
Chronic, *n* (%)	48 (22.5)	7 (8.5)	41 (31.3)	
Preoperative complications				
Cerebrovascular disease, *n* (%)	25 (11.7)	11 (13.4)	14 (10.7)	0.547
Coronary artery disease, *n* (%)	44 (20.7)	22 (26.8)	22 (16.8)	0.078
CKD stage ≥ 4, *n* (%)	30 (14.1)	15 (18.2)	15 (11.5)	0.163
COPD, *n* (%)	19 (8.9)	12 (14.6)	7 (5.3)	0.021
EF ≤ 40%, *n* (%)	1 (0.5)	1 (1.2)	0	0.385
Previous median sternotomy, *n* (%)	11 (5.2)	6 (7.3)	5 (3.8)	0.342
Emergency, *n* (%)	33 (15.5)	11 (13.4)	22 (16.8)	0.507
Logistic EuroSCORE (%)	20.9 ± 14.8	23.3 ± 16.1	19.3 ± 12.4	0.045
Preoperative measurements				
Maximum aortic diameter (mm)	51 (44–60)	56 (48–64)	50 (42–58)	<0.001
Diameter of proximal LZ (mm)	31.3 ± 3.5	32.4 ± 3.1	30.7 ± 3.6	0.001
Length of proximal LZ (mm)	20.0 (16.9–24.0)	20.0 (16.5–25.0)	19.0 (17.0–24.0)	0.462
Diameter of distal LZ (mm)	27.1 ± 4.2	27.7 ± 4.4	26.8 ± 4.1	0.135

Data are represented as mean ± standard deviation and median (IQR: interquartile range). TAA: thoracic aortic aneurysm; DAA: dissecting aortic aneurysm; CKD: chronic kidney disease; COPD: chronic obstructive pulmonary disease; EF: ejection fraction; LZ: landing zone.

**Table 2 jcm-12-05326-t002:** Procedure and stent grafts.

	All *n* = 213	Zone 1 *n* = 82 (38.5%)	Zone 2 *n* = 131 (61.5%)	*p* Value
Procedure success, *n* (%)	213 (100)	82 (100)	131 (100)	1.00
Operative time (minutes)	160 (130–204)	191 (161–220)	140 (112–171)	<0.001
Stent grafts				
Type of proximal stent grafts				
TAG and CTAG, *n* (%)	162 (76.1)	65 (79.3)	97 (74.1)	
Zenith TX2 and TXD, *n* (%)	17 (8.0)	7 (8.5)	10 (7.6)	
Relay Plus and NBS, *n* (%)	26 (12.2)	8 (9.8)	18 (13.7)	
Talent and Valiant, *n* (%)	8 (3.8)	2 (2.4)	6 (4.6)	
Number of stent grafts (*n*)	1.4 ± 0.6	1.5 ± 0.5	1.5 ± 0.6	0.822
Proximal stent graft				
Diameter (mm)	36 (34–38)	37 (34–40)	34 (34–37)	0.004
Oversize rate (%)	14 (10–18)	13 (10–17)	14 (10–19)	0.154
Distal stent graft				
Diameter (mm)	34 (30–37)	34 (31–37)	34 (28–35)	0.021
Oversize rate (%)	19 (14–28)	20 (14–29)	19 (13–28)	0.275
Th of distal stent graft edge	8 (7–9)	8 (7–8)	8 (7–9)	0.256

Data are represented as mean ± standard deviation and median (IQR: interquartile range). Th: thoracic spine.

**Table 3 jcm-12-05326-t003:** In-hospital outcomes.

	All *n* = 213	Zone 1 *n* = 82 (38.5%)	Zone 2 *n* = 131 (61.5%)	*p* Value
Hospital stay (days)	10 (8–16)	11 (9–19)	10 (8–14)	0.005
Discharge at home, *n* (%)	203 (95.3)	78 (95.1)	125 (95.4)	1.00
30-day mortality, *n* (%)	1 (0.5)	1 (1.2) *	0	0.385
Abdominal embolic event, *n* (%)	1 (0.5)	1 (1.2) *	0	0.385
Hospital mortality, *n* (%)	2 (0.9)	1 (1.2) *	1 (0.8) ^+^	1.00
Abdominal embolic event, *n* (%)	1 (0.5)	1 (1.2) *	0	0.385
Aneurysm rupture, *n* (%)	1 (0.5)	0	1 (0.8) ^+^	1.00
Aortic events				
Complications				
Stroke, *n* (%)	2 (0.9)	1 (1.2)	1 (0.8)	1.00
Spinal cord injury, *n* (%)	2 (0.9)	1 (1.2)	1 (0.8)	1.00
Cardiac events, *n* (%)	0	0	0	1.00
Abdominal embolic event, *n* (%)	1 (0.5)	1 (1.2) *	0	0.385
New dialysis, *n* (%)	0	0	0	1.00
Aneurysm rupture, *n* (%)	1 (0.5)	0	1 (0.8) ^+^	1.00
RTAD, *n* (%)	0	0	0	1.00
Distal SINE, *n* (%)	0	0	0	1.00
Stent graft infection, *n* (%)	0	0	0	1.00
Bypass graft occlusion, *n* (%)	0	0	0	1.00
Endoleaks				
Type 1a, *n* (%)	3 (1.4)	1 (1.2)	2 (1.5) ^+^	1.00
Type 1b, *n* (%)	1 (0.5)	1 (1.2)	0	0.385
Type 1c, *n* (%)	7 (3.3)	3 (3.7)	4 (3.1)	1.00
Type 2, *n* (%)	0	0	0	1.00
Type 3, *n* (%)	0	0	0	1.00

Data are represented as median (IQR: interquartile range). RTAD: retrograde type A dissection; SINE: stent graft-induced new entry; *: same patient; ^+^: same patient.

**Table 4 jcm-12-05326-t004:** Late outcomes.

	All *n* = 213	Zone 1 *n* = 82 (38.5%)	Zone 2 *n* = 131 (61.5%)	*p* Value
Late death, *n* (%)	38 (17.8)	19 (23.2)	19 (14.5)	
Stent graft infection, *n* (%)	1 (0.5)	1 (1.2)	0	0.385
Cardiac events, *n* (%)	3 (1.4)	1 (1.2)	2 (1.5)	1.00
Cerebrovascular diseases, *n* (%)	6 (2.8)	2 (2.5)	4 (3.1)	1.00
Malignancy, *n* (%)	10 (4.7)	7 (8.5)	4 (3.1)	0.079
Infection, *n* (%)	17 (8.0)	7 (8.5)	9 (6.9)	0.654
Others, *n* (%)	1 (1.2)	1 (1.2)	0	0.385
Aortic events				
Complications				
Stroke, *n* (%)	0	0	0	1.00
Spinal cord injury, *n* (%)	0	0	0	1.00
Cardiac events (%)	0	0	0	1.00
Abdominal embolic event, *n* (%)	0	0	0	1.00
New dialysis, *n* (%)	0	0	0	1.00
Aneurysm rupture, *n* (%)	0	0	0	1.00
RTAD, *n* (%)	1 (0.5)	1 (1.2)	0	0.385
Distal SINE, *n* (%)	3 (1.4)	0	3 (2.3)	0.286
Stent graft infection, *n* (%)	1 (0.5)	1 (1.2)	0	0.385
Bypass graft occlusion, *n* (%)	1 (0.5)	0	1 (0.8)	1.00
Endoleaks				
Type 1a, *n* (%)	4 (1.9)	1 (1.2)	3 (2.3)	1.00
Type 1b, *n* (%)	2 (0.9)	2 (2.5)	0	0.147
Type 1c, *n* (%)	0	0	0	1.00
Type 2, *n* (%)	0	0	0	1.00
Type 3, *n* (%)	0	0	0	1.00

RTAD: retrograde type A dissection; SINE: stent graft-induced new entry.

## Data Availability

Data cannot be shared for ethical/privacy reasons. The data underlying this article cannot be shared publicly due to the privacy of individuals that participated in the study. On reasonable request, the data will be made available by the corresponding author after approval from the Ethical Committee of the University of Osaka.

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
