# Peer review of "Assessment of the Effectiveness of Zone 1-Landing Hybrid TEVAR by Comparing Its Outcomes with Those of Zone 2-Landing Hybrid TEVAR"

_jcm, 2023, doi:10.3390/jcm12165326_

Round 1

Reviewer 1 Report

The present manuscript reports interesting data regarding hybrid TEVAR.

First of all please tranfer tables and figures of secondary outcomes in the suplemmentary materials.

Minor editing is required.

Author Response

  1. First of all, please transfer tables and figures of secondary outcomes in the supplementary materials.

ANSWER: Thank you for your thoughtful comment. We moved Table 5 and Figure 4 to supplementary material.

CHANGES: We moved Table 5 and Figure 4 to supplementary material.

Reviewer 2 Report

Dear Editor, 

I read with interest the article by Tomoaki Kudo et al. regarding the effectiveness of zone 1 landing hybrid TEVAR by comparing its outcomes with those of zone 2 landing hybrid TEVAR. 

The authors concluded that zone 1 and 2 landings hybrid TEVAR outcomes were satisfactory. Despite the high procedural difficulty and surgical risk, the outcomes of zone 1 landing hybrid TEVAR were equal to that of zone 2 landing hybrid TEVAR. If the surgical risk is high, zone 1 landing hybrid TEVAR should not be avoided.

The manuscript is well written and of interest, with a large number of patients included in the study. Also, the design of the study as well as the materials and methods section are clearly presented and easy to follow. Very well-presented results. For such an article, the authors must improve the number of references (25 are very few).

Author Response

  1. The manuscript is well written and of interest, with a large number of patients included in the study. Also, the design of the study as well as the materials and methods section are clearly presented and easy to follow. Very well-presented results. For such an article, the authors must improve the number of references (25 are very few).

ANSWER: Thank you for your thoughtful comment. We added the references.

CHANGES: We added the references.

Reviewer 3 Report

Thank you for conducting this study on effectiveness of zone 1 and 2 landing during hybrid TEVAR. However, there are several issues that need to be addressed. My major concern is that the data presented in this article appear to have already been published in another journal, with a different analysis. 

[Reference: Kudo T, Kuratani T, Sawa Y, Miyagawa S. Effectiveness of Proximal Landing Zone 1 and 2 Thoracic Endovascular Aortic Repair for Type B Aortic Dissection by Comparing Outcomes With Thoracic Arch Aneurysm. J Endovasc Ther. 2023 May 24:15266028231174407. doi: 10.1177/15266028231174407. Epub ahead of print. PMID: 37222467.]

I have provided specific comments below to highlight areas of concern and potential improvements:

1.     The study focuses solely on one hybrid surgical approach without considering other surgical modalities, such as total open or total endo arch repairs with fenestrated and inner branched stentgrafts. Including a discussion of these alternative approaches could enrich the article's content.

2.     The authors mentioned zero losses during follow-up (FU), which is impressive. However, it would be valuable to know how this was achieved and whether the cause of deaths was cross-checked with any national death registries. If this was done, please provide further details.

3.     In Figure 1(B), the use of long axillo-axillo bypass for LSA debranching is discussed. Clarifying the rationale behind this choice, especially when a short carotid-subclavian bypass is more commonly used, would be beneficial.

4.     Please specify the number of patients with connective tissue disorders included in the study, if any.

5.     It is essential to state whether the same oversizing pattern was followed for both aneurysms and dissections. Additional technical notes or remarks related to the debranching method and TEVAR procedures should be included to provide more comprehensive insights.

6.     Clarify whether the hybrid procedures were performed in a single session or multiple sessions. Provide a rationale for the chosen approach and any special considerations associated with each.

7.     The reason for the tight follow-up schedule mentioned on page 3, line 85, needs to be explained.

8.     Consider adding "technical success" as an outcome measure in the study.

9.     There is no information regarding techniques employed to prevent spinal cord ischemia (SCI). Include details about the protocol followed, such as prophylactic or therapeutic spinal drainage and other neuro-monitoring during procedures.

10.  The low incidence of stroke reported in both groups is noteworthy. Provide a clear definition of stroke and elaborate on the neurological assessment performed. Mention whether CT or MR scans were used to verify stroke occurrences.

11.  The article contains an extensive number of Kaplan-Meier figures. To enhance clarity, consider summarizing only a few in the main text and moving the others to supplementary material. Figures 2B, 2C, 3B, 3C and all 4 could just added as supplementary material. Instead of Figure 4 only one KM comparing survival of TAA vs DAA could be considered. 

12.  Acknowledge potential selection bias as a limitation in the appropriate section of the article, as is common for studies of this nature. A comparative study between different treatment modalities would be valuable and should be considered.

Sufficient though moderate editing is needed.

Author Response

  1. The authors mentioned zero losses during follow-up (FU), which is impressive. However, it would be valuable to know how this was achieved and whether the cause of deaths was cross-checked with any national death registries. If this was done, please provide further details.

ANSWER: Thank you for your thoughtful comment.

CHANGES: We achieved no lost follow because we did a strict follow-up. We did not cross-check with any national death registries.

  1. In Figure 1(B), the use of a long axilla-axilla bypass for LSA debranching is discussed. Clarifying the rationale behind this choice, especially when a short carotid-subclavian bypass is more commonly used, would be beneficial.

ANSWER: Thank you for your thoughtful comment. As you pointed out, we think that the shorter bypass graft is better. However, we try not to do the cervical procedure to prevent cerebral infarction, if possible.

  1. Please specify the number of patients with connective tissue disorders included in the study, if any.

ANSWER: Thank you for your thoughtful comment. There were no patients with connective tissue disorders in the study.

  1. It is essential to state whether the same oversizing pattern was followed for both aneurysms and dissections. Additional technical notes or remarks related to the debranching method and TEVAR procedures should be included to provide more comprehensive insights.

ANSWER: Thank you for your thoughtful comment. We added the sentence.

CHANGES: We added the sentence, “We selected the stent graft with the same oversizing for aneurysm and dissection.”

  1. Clarify whether the hybrid procedures were performed in a single session or multiple sessions. Provide a rationale for the chosen approach and any special considerations associated with each.

ANSWER: Thank you for your thoughtful comment. We added the sentence.

CHANGES: We added the sentence, “TEVAR was performed after the bypass within the same surgery.”

  1. The reason for the tight follow-up schedule mentioned on page 3, line 85, needs to be explained.

ANSWER: Thank you for your thoughtful comment. The follow-up schedule for this study is normal for us.

  1. Consider adding "technical success" as an outcome measure in the study.

ANSWER: Thank you for your thoughtful comment. We mentioned that all procedures were successful in the manuscript.

  1. There is no information regarding techniques employed to prevent spinal cord ischemia (SCI). Include details about the protocol followed, such as prophylactic or therapeutic spinal drainage and other neuro-monitoring during procedures.

ANSWER: Thank you for your thoughtful comment. We added the sentence.

CHANGES: We added the sentence, “Cerebrospinal fluid drainage was performed on patients with stent graft distal edge Th≤10.”

  1. The low incidence of stroke reported in both groups is noteworthy. Provide a clear definition of stroke and elaborate on the neurological assessment performed. Mention whether CT or MR scans were used to verify stroke occurrences.

ANSWER: Thank you for your thoughtful comment. We did not routinely perform CT or MRI after the procedure. We added the sentence.

CHANGES: We added the sentence, “We defined a stroke as having neurologic symptoms.”

  1. The article contains an extensive number of Kaplan-Meier figures. To enhance clarity, consider summarizing only a few in the main text and moving the others to supplementary material. Figures 2B, 2C, 3B, 3C and all 4 could just added as supplementary material. Instead of Figure 4 only one KM comparing survival of TAA vs DAA could be considered.

ANSWER: Thank you for your thoughtful comment. We moved Table 5 and Figure 4 to supplementary material.

CHANGES: We moved Table 5 and Figure 4 to supplementary material.

Round 2

Reviewer 3 Report

Thank you. No further comments.

A thorough reading and language editing is needed.